# Broadband Continuous Transverse Stub (CTS) Array Antenna for High-Power Applications

**DOI:** 10.3390/mi14112127

**Published:** 2023-11-20

**Authors:** Yunfei Sun, Kelin Zhou, Juntao He, Zihan Yang, Chengwei Yuan, Qiang Zhang

**Affiliations:** College of Advanced Interdisciplinary Studies, National University of Defense Technology, Changsha 410073, China; sunyunfei_gfkd@163.com (Y.S.); zhoukelin98@163.com (K.Z.); hejuntao12@163.com (J.H.); 15234534500@163.com (Z.Y.); cwyuan@nudt.edu.cn (C.Y.)

**Keywords:** continuous transverse stub (CTS) array, electromagnetic band gap (EBG), broadband antenna, high-power microwave (HPM)

## Abstract

A continuous transverse stub (CTS) array antenna with broad bandwidth and high-power handling capacity is proposed in this paper. The technologies of multi-step impedance matching and T-shaped electromagnetic band-gap (EBG) loading are utilized, which improved the antenna operating frequency bandwidth. An H-plane lens horn is used to feed the CTS array. As a result, a good bandwidth capability of more than 32% is achieved, with a gain variation less than 3.0 dB. The measured sidelobe level (SLL) is below −18 dB in the entire frequency range. Moreover, the power handling capacity of the antenna is more than 80 MW and can reach the GW level after arraying, which indicates that this antenna has application potential in the high-power microwave (HPM) field.

## 1. Introduction

High-power microwave (HPM) refers to electromagnetic waves with operating frequencies between 1 and 300 GHz and peak power exceeding 100 MW [1,2,3]. HPM technology emerged in the 1970s as a product of the combination of pulsed power technology, plasma physics, and electric vacuum technology. Due to the limitations of the physical mechanisms that generate HPMs, existing HPM sources are usually narrow-band with bandwidth of only MHz to tens of MHz. Radiation systems play a critical role as they determine whether the HPM energy can be utilized efficiently, and the existing radiation system bandwidth is also narrow. In our previous research, we designed many types of beam-steering antennas with the transmit array (T-A) lens [4,5,6,7] for HPM application and achieved a beam-steering range of 91.2° in the conical region. Compared with the existing HPM beam-steering technology, this method is simple and convenient, dispensing with feed networks. At the same time, many novel T-A structures were introduced into the HPM range, and they greatly promoted the HPM antenna technology. However, the bandwidth of the beam-steering lens antenna is narrow. With the expansion of HPM applications [8,9,10,11], HPM sources with a single frequency can no longer meet the requirements. The study of frequency-tunable HPM sources has become an important direction for HPM technology [12,13,14,15,16,17,18,19,20]. Frequency-tunable HPM sources have two advantages: (1) they can adjust the output frequency according to the different types of targets, thereby increasing the coupling probability between HPM and targets and enhancing the effectiveness of HPM on targets; (2) in the study of the HPM effect, by changing the frequency of the output, the microwave can improve the efficiency of the effect in experiments and help us to understand the physical mechanism of the interaction between HPM and the target. At present, the frequency-tunable HPM source mainly includes Gyrotron, relativistic magnetron (RM), relativistic backward wave oscillator (RBWO), transit-time oscillator (TTO), magnetically insulated transmission line oscillator (MILO), and the tuning bandwidth could reach about 30%. However, the existing HPM antennas, such as mode conversion antenna, radial line helical array antenna, waveguide slotted array antenna, and radial line slotted array antenna [21,22,23,24,25,26], have a narrow bandwidth, which cannot meet the demand for frequency-tunable HPM sources. As a result, an HPM antenna that can realize broadband radiation is an urgent need.

Continuous transverse stub (CTS) technology is widely utilized in satellite and mobile wireless communications owing to the merits of low loss, wide bandwidth, dimensional robustness, design flexibility, and fabrication stability. The reported CTS arrays are usually fed by coplanar waveguides, coaxial waveguides, or parallel plate waveguides, in which the parallel plate waveguide CTS antenna-adopted all-metal structure may have high-power handling capacity and wide bandwidth [27,28,29,30,31,32]. At the same time, the antenna can also have good conformal characteristics and beam-scanning ability [33,34,35,36,37]. In light of the above advantages, the CTS array has broad application prospects in the HPM field. In this paper, a CTS array antenna with broadband radiation capability is proposed. An HPM radiation system operating in the X-band is designed and measured. After adding a T-shaped electromagnetic band-gap (EBG) structure on the CTS surface, the relative operating bandwidth of the antenna could reach 32%. Moreover, the power handling capacity of the antenna is more than 80 MW and can reach the GW level after arraying, which provides an option for frequency-tunable HPM source radiation.

## 2. Materials and Methods

This section discusses the antenna unit, array design process, and antenna feeding, respectively. The overall structure of the antenna is shown in Figure 1. The antenna consists of two parts, where part 1 is the feed lens horn and part 2 is the CTS antenna array. The antenna size is 1230 mm × 160 mm × 58 mm.

### 2.1. Antenna Element

The structure of a non-uniform four-step CTS element loading a T-shaped EBG structure is shown in Figure 2, where *s*_1_ to *s*_4_ represent the width of the four steps and *h*_1_ to *h*_4_ represent the height of the four steps, respectively. *s* and *t* are the width and height of the EBG roof, respectively, and *h*_5_ is the height of the EBG root. *d* is the interval of the adjacent slots, and *l* is the width of the slot. *b* is the height of the parallel flat plate and is equal to 10.16 mm. The T-shape structure is used to suppress the surface-wave reflection at low frequencies. The magnetic boundary is set on the left and right sides to simulate the element. The open add space boundary is set on the top side of the element. The calculation accuracy is set to −40 dB, and adaptive mesh refinement is used. The final number of mesh cells of the element is about 87,000. The unit is small, so the number of mesh cells can ensure that the calculation is accurate. The values of each parameter are defined as shown in Table 1.

The unit coupling coefficient *K*^2^ is defined as
(1)$$K^{2}=1-S_{11}^{2}-S_{21}^{2}$$

The ability of the CTS element to couple microwaves outward through the slot is characterized. According to the theoretical derivation, the maximum value of *K*^2^ is 0.5.

The coupling ability of this element is compared with that of a non-uniform four-step element and a uniform four-step element at 10 GHz, and the results are shown in Figure 3. The red line represents the non-uniform CTS, the black line represents the uniform CTS, and the blue line represents the CTS-loaded T-shaped EBG. It can be seen that when the slot width *l* is small, the coupling ability of the element with T-shaped EBG is significantly improved, which means that the structure can radiate more power into free space at the same size.

### 2.2. Antenna Array

The resistance of a CTS in the periodical circumstance can be obtained by measuring or numerically calculating the *S*_21_ and *S*_11_ of the slotted waveguide with *N* uniform slots [18],
(2)$$r={\left(\frac{{\left|S_{21}\right|}^{2}}{1-{\left|S_{11}\right|}^{2}}\right)}^{-1/N}-1$$

According to Equation (2), the relationship of the slot equivalent resistance and the slot width can be obtained using simulation software, such as Computer Simulation Technology (CST) Microwave Studio 2010 and data fitting. Figure 4 shows the relationship between the slot width and the equivalent resistance when the parallel flat plate height is 10.16 mm. The fitting formulas are expressed by Equation (3).
(3)$$l=0.7848+23.8176r-51.9595r^{2}+43.0419r^{3}+70.1345r^{4}$$

Taking into account the power handling capacity, antenna efficiency, and the range of the equivalent resistance, we choose a uniform aperture distribution to design the antenna. The radiation efficiency of the antenna is set as 98%. The total number of slots is 39. Since the previous simulation was established on both sides of the antenna as an ideal magnetic boundary, it can be considered as a parallel flat waveguide with TEM mode as its fundamental mode, while in practical application, the two sides are closed electrical walls and the fundamental mode inside the waveguide is TE_n0_, so the antenna width will certainly affect the antenna performance. According to the simulation, when *a* = 121.5 mm, the electric field transmitted in the waveguide can be considered as quasi-TEM mode. So, we choose *a* = 121.5 mm in the following design and simulation.

The standing wave array can only operate at a specific frequency, with slots spaced at half a waveguide wavelength, and its bandwidth is narrow. At the same time, the main lobe is perpendicular to the antenna surface. To obtain a wide bandwidth, the antenna must be designed as a traveling wave array, with a slot interval less than half a waveguide wavelength, and the beam direction of the antenna will deviate from the normal direction. The simulated results are shown in Figure 5 and Figure 6. In the frequency range of 8.4–11.6 GHz, the reflection coefficient *S*_11_ is less than −20 dB, and the remaining energy (*S*_21_) at the waveguide end is less than −20 dB. The metal is aluminum, the estimated ohmic loss is less than 1%, and the antenna efficiency can reach 98%. In the whole frequency band, the gain variation is less than 2.8 dB, the sidelobe level is less than −19 dB, and the antenna relative bandwidth is 32%. Figure 7 shows the 3D radiation pattern at 10 GHz, where the gain is 30.1 dB and the sidelobe is −19.1 dB. These results indicate the antenna has a good performance and wide bandwidth, which can meet the requirements of the existing frequency-tunable HPM source.

After simulation, the internal electric field of the antenna is shown in Figure 8 at 8.4 GHz. Before loading the T-shaped EBG structure, the electric field is bound to the antenna surface and propagated to the end. It is a typical surface-wave characteristic. After the T-shaped EBG structure is loaded, the internal electric field of the antenna has a clear tilted straight line, and the electromagnetic wave radiates towards free space, indicating that the antenna surface wave is fully suppressed at low frequency.

### 2.3. Antenna Feeding

In order to feed the antenna with a width of 121.5 mm and a height of 10.16 mm, an H-plane lens horn was designed. The horn is fed by a standard rectangular waveguide BJ100 made by Xi’an Hengda Microwave Technology Development Company, China, and the working frequency is 8.2–12 GHz, which can meet the working frequency band of the CTS antenna. The width of the horn surface is 121.5 mm, the height is 10.16 mm, and the horn transition section is 142.45 mm. The lens is made of polyethylene with a dielectric constant ε = 2.3. To reduce the reflection coefficient, adding a dielectric matching layer for impedance matching is necessary. The focal length is f = 154.5 mm and thickness is d = 19.9 mm of the dielectric lens. The matching layer on both sides is made of alloy blend composite (ABC) material, with a dielectric constant ε = 1.5. The thickness of the matching layer on the bending section is 5.6 mm, and the thickness on the vertical end is 6.7 mm.

The simulation results of the horn are shown in Figure 9. It can be seen that the reflection S_11_ is less than −20 dB in the whole frequency band, and the high-order mode is suppressed to a certain extent. The transmission coefficient of high-order mode *S*_2(3)1(1)_ is less than −15 dB, and *S*_2(5)1(1)_ is less than −20 dB. In addition, 40 mm from the aperture surface, the phase difference is within 20°, which can be considered to form an approximate plane wave. The electric field distribution inside the horn is shown in Figure 9c. It can be seen that the spherical wave is transformed into a plane wave on the horn surface. At 10 GHz, the injection power is 0.5 W, the maximum electric field inside the horn is 2148 V/m, and the maximum electric field in the dielectric lens is 900 V/m. Based on the vacuum breakdown field strength of 50 MV/m of the metal and 4 MV/m of the dielectric [38,39], the estimated power capacity is about 95 MW.

## 3. Results

This section processes the CTS antenna and H-plane horn for testing. Low-power (*S* parameter and radiation pattern measurement) and high-power tests are carried out, respectively.

### 3.1. Low-Power Test

As shown in Figure 10, a vector network analyzer (VNA) is used to measure the reflection coefficient *S*_11_. A waveguide coaxial converter is used to connect the VNA and the H-plane horn. An absorbing material is placed at the antenna end to absorb the remaining energy. The measured and simulated reflection coefficients are shown in Figure 11. It can be seen that several resonant points appear in the reflection curves, and the overall reflection level is consistent with the simulation results. During the processing, due to limitations in the bonding process, the two materials cannot be completely bonded together, which may cause the difference between the simulation result and the measured data. In addition, the reflection coefficient is below −10 dB in the range of 8.4–11.6 GHz.

The length of the antenna is 1 m, and the corresponding far-field distance is more than 66 m. We do not have a very large anechoic chamber, so the radiation pattern of the antenna was measured using a near-field testing system. The measurement environment is shown in Figure 12, and the measurement frequency ranges from 8.4 to 11.6 GHz with an interval of 0.4 GHz. The probe is 5 *λ*_0_ away from the antenna surface. In this experiment, the sampling range along the horizontal direction is 1200 mm, the sampling interval is 10 mm, and there are 120 sampling points; the sampling range along the vertical direction is 400 mm, with a sampling interval of 10 mm and 40 sampling points. Figure 13 shows the radiation pattern measured at 8.4 GHz, and Figure 14 shows the radiation pattern measured at 10 GHz. The measured results are in good agreement with the simulated results, and the cross-polarization is much lower than the co-polarization, which proves that the antenna radiates a linearly polarized wave.

Figure 15 shows the radiation pattern measured at 8.4–11.6 GHz. It can be seen that the antenna radiation performance is good in the whole band. Table 2 gives the measured results in typical frequency, where the maximum gain is 31.1 dB, the minimum gain is 28.2 dB, and the gain variation is 2.9 dB. The sidelobe level is less than −18 dB, and the width of the main lobe is within 2.5°. The working bandwidth of the antenna can reach 32%.

### 3.2. High-Power Microwave Test

To identify the power handling capacity of the designed antenna, the high-power test was carried out by using a transit-time oscillator (TTO) in our laboratory. The maximum output power of the HPM source is 800 MW, with a frequency of 9.44 GHz and a pulse width of 38 ns. All of our measurements were carried out under this condition. A mechanical pump and a molecular pump were used to realize a 4 × 10^−3^ Pa vacuum degree. Since the output power far exceeds the antenna’s power handling capacity, to avoid RF breakdown, a coupler with a coupling coefficient of −10 dB was designed and used. The structure of the coupler is shown in Figure 16, and the *S* parameters are shown in Figure 17. In the 9.3~9.5 GHz frequency band, the reflection coefficient *S*_2(3)1(3)_ is below −20 dB, and the coupling coefficient *S*_3(1)1(3)_ is −10 dB, which can meet the experimental requirements.

As shown in Figure 18, when the HPM passes through the coupler, 10 percent power will inject into the CTS antenna, and the remaining power will radiate through the horn. The microwave radiated by the CTS antenna and the horn will be received by two horn antennas located in the far-field region. Moreover, another coupler which the coupling coefficient can adjust through a change in the insertion depth of a metal rod is designed. It can achieve a wide range of coupling coefficient adjustments and will be used in the future.

The typical measured output waveform is shown in Figure 19. The pulse width of the microwave radiated by the horn is 37.2 ns. The pulse width of the microwave radiated by the CTS antenna is 32 ns, which has no obvious pulse width shortening compared to the horn. Therefore, it can be judged that no electric field breakdown occurs inside the CTS antenna. The power handling capacity of the CTS antenna can reach 80 MW.

## 4. Discussion

In the experiments, the power handling capacity of the antenna only reached 80 MW; the limitation of the power handling capacity is the horn. To further improve the power handling capacity of the antenna, there are two solutions. The first solution is to use a pillbox box to feed the antenna, as shown in Figure 20. The pillbox box is divided into two layers, where the lower layer is fed by a small-angle horn. The input of the horn is a BJ100 rectangular waveguide. When the spherical wave generated by the small angle horn is transmitted to the parabolic wall, it is reflected and coupled to the upper layer. Due to the coincidence of the focus of the parabolic and the phase center of the horn, it can produce a plane wave and feed the CTS antenna. In this scheme, the size of the pillbox box is larger, so it can withstand higher handling power capacity.

Another solution is to use a network to feed the CTS antenna, as shown in Figure 21. In this scheme, the microwave is fed into the TEM mode through a coaxial waveguide, converted into radial transmission through a transition cone, and then evenly divided into N-way output TE_10_ modes to feed the CTS antenna. The output number of the power divider can be designed as needed, and it is easy to achieve GW-level power handling capacity.

## 5. Conclusions

In this paper, an HPM CTS antenna with a wide band was designed. The non-uniform four-step structure was adopted to enhance the coupling ability of antenna elements. By adding the T-shaped EBG structure, the working bandwidth was extended. To feed the antenna, a horn with a dielectric lens was designed, and the bandwidth of the horn can meet the requirements. To verify the performance of the CTS antenna, low-power and high-power experiments were carried out. The results show that the antenna has good radiation performance at 8.4–11.6 GHz, with a working bandwidth of 32%, and has a power handling capacity over 80 MW. These results indicate that this antenna can be used for tunable HPM source radiation.

## Figures and Tables

**Figure 1 micromachines-14-02127-f001:**
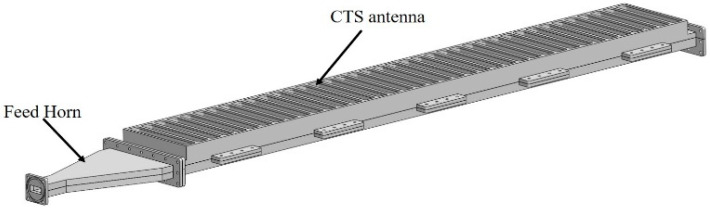
Three-dimensional structure of non-uniform four-step CTS antenna.

**Figure 2 micromachines-14-02127-f002:**
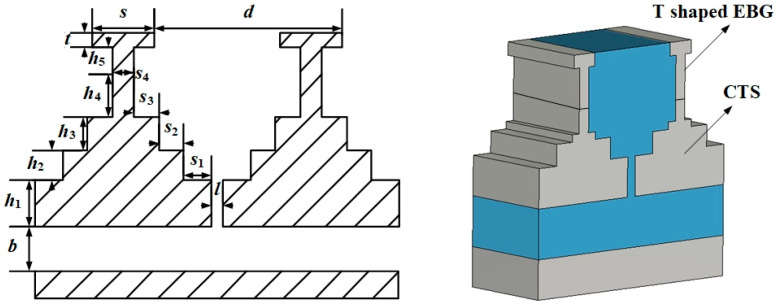
The structure of non-uniform four-step CTS element.

**Figure 3 micromachines-14-02127-f003:**
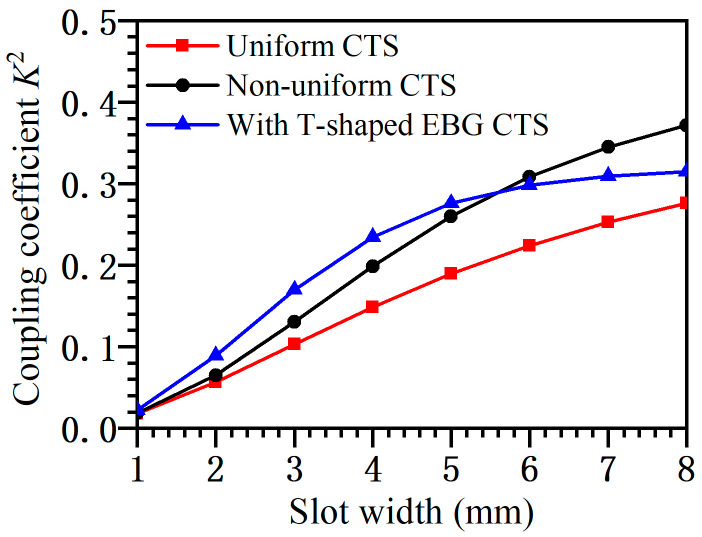
Coupling coefficient of the three structures.

**Figure 4 micromachines-14-02127-f004:**
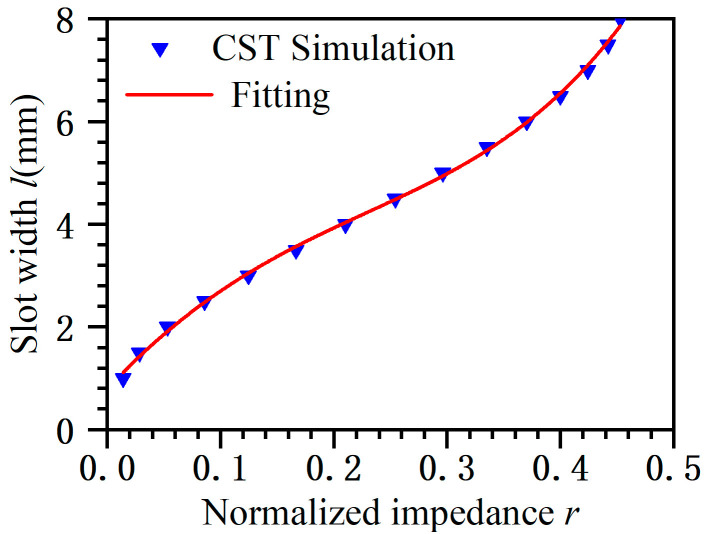
Data fitting and simulation comparison.

**Figure 5 micromachines-14-02127-f005:**
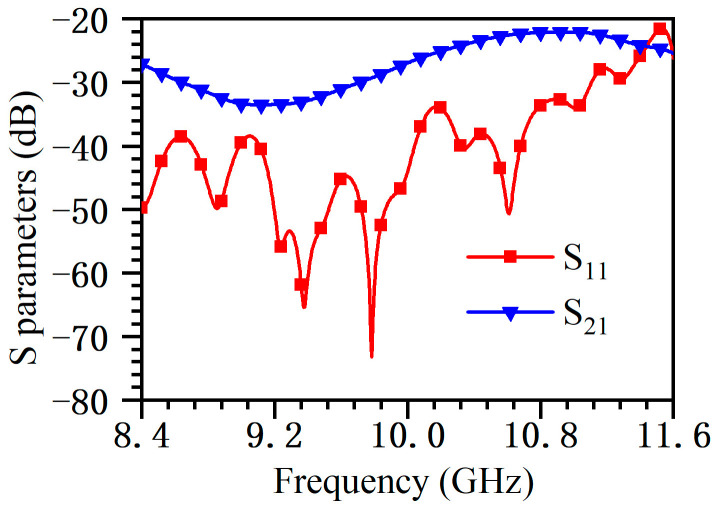
*S* parameters.

**Figure 6 micromachines-14-02127-f006:**
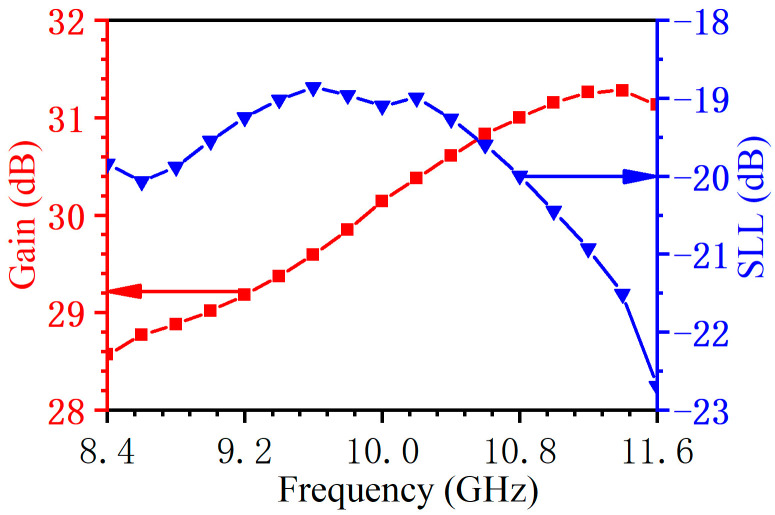
Simulated gain and sidelobe.

**Figure 7 micromachines-14-02127-f007:**
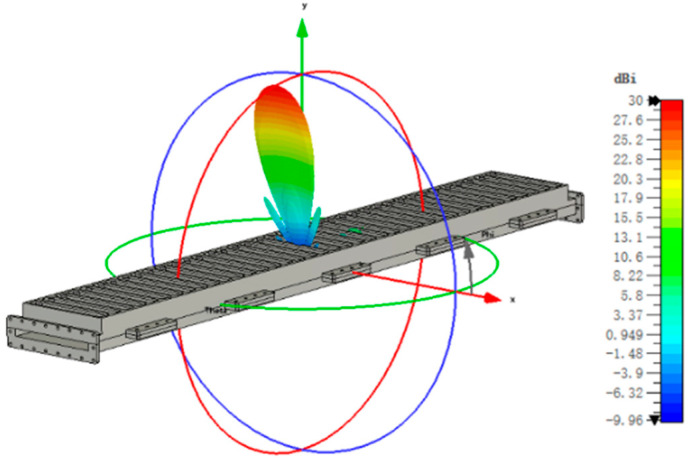
Antenna 3D radiation pattern at 10 GHz.

**Figure 8 micromachines-14-02127-f008:**
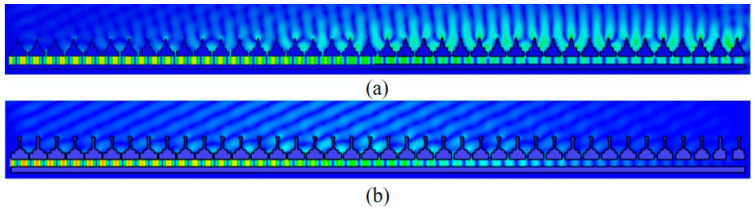
Electric field inside the antenna: (**a**) without T-shaped EBG, (**b**) with T-shaped EBG.

**Figure 9 micromachines-14-02127-f009:**
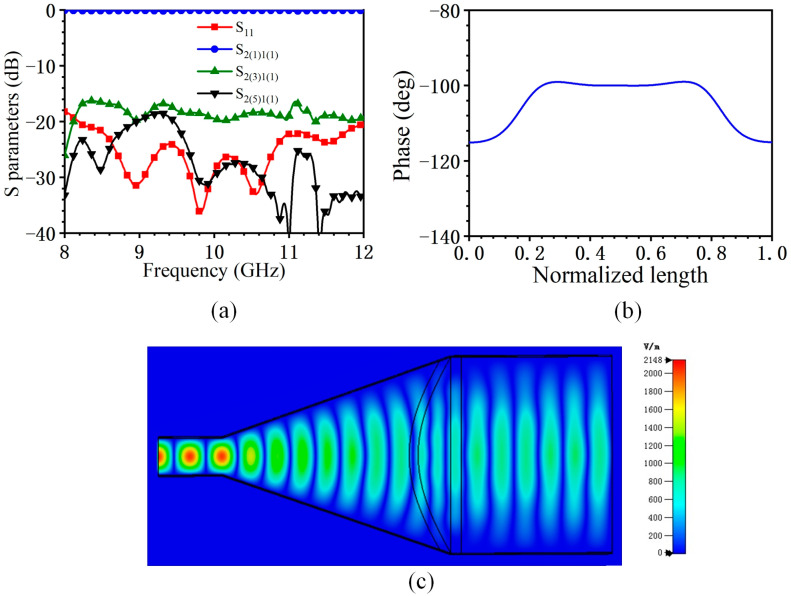
Simulated results of the feed horn. (**a**) *S* parameters of the horn. (**b**) Phase distribution of electric field on flare surface. (**c**) Electric field distribution.

**Figure 10 micromachines-14-02127-f010:**
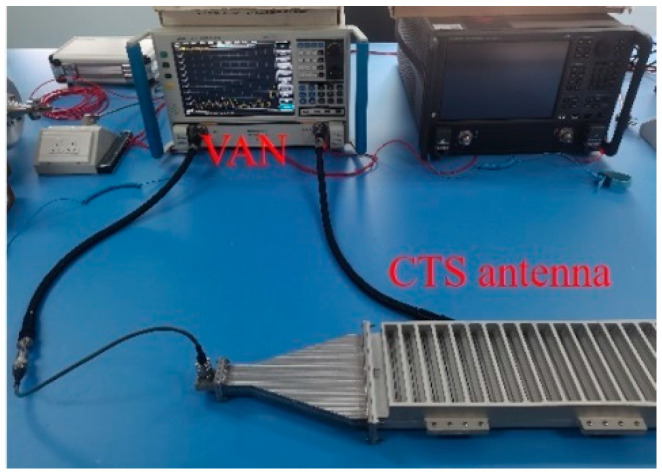
Measurement of S_11_.

**Figure 11 micromachines-14-02127-f011:**
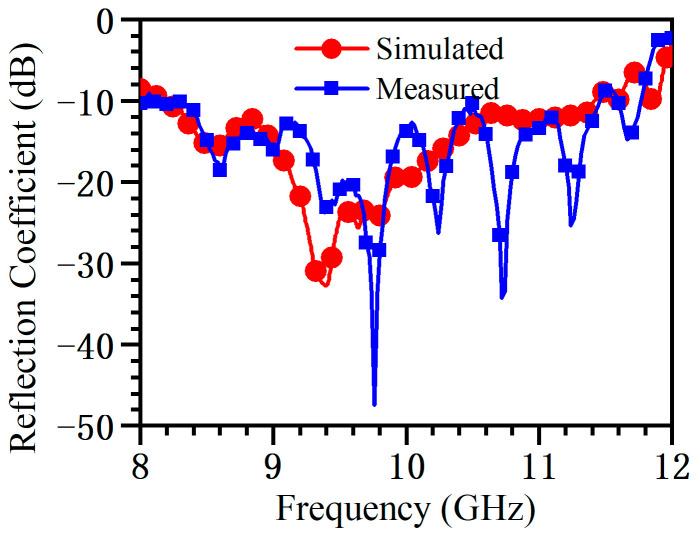
Reflection coefficient for antenna simulation and measurement.

**Figure 12 micromachines-14-02127-f012:**
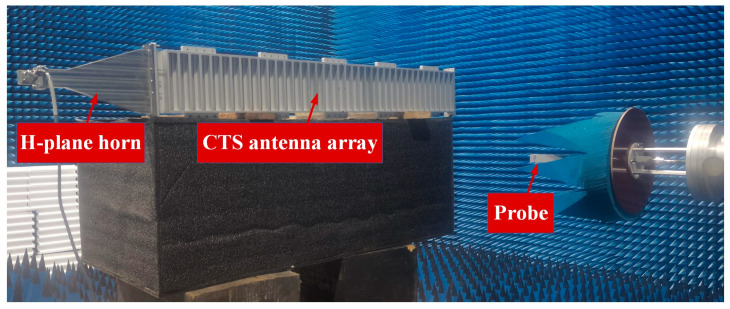
Measurement environment of the antenna pattern.

**Figure 13 micromachines-14-02127-f013:**
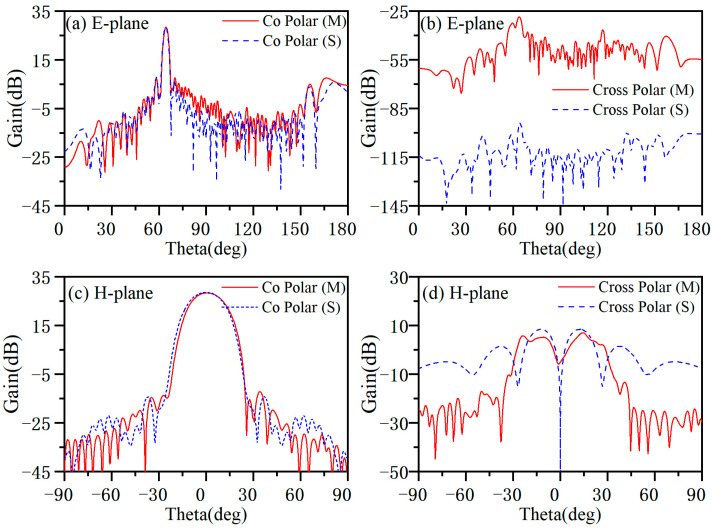
Measured and simulated radiation pattern of the antenna at 8.4 GHz. (**a**) Co-polarization radiation pattern of E-plane. (**b**) Cross-polarization radiation pattern of E-plane. (**c**) Co-polarization radiation pattern of H-plane. (**d**) Cross-polarization radiation pattern of H-plane.

**Figure 14 micromachines-14-02127-f014:**
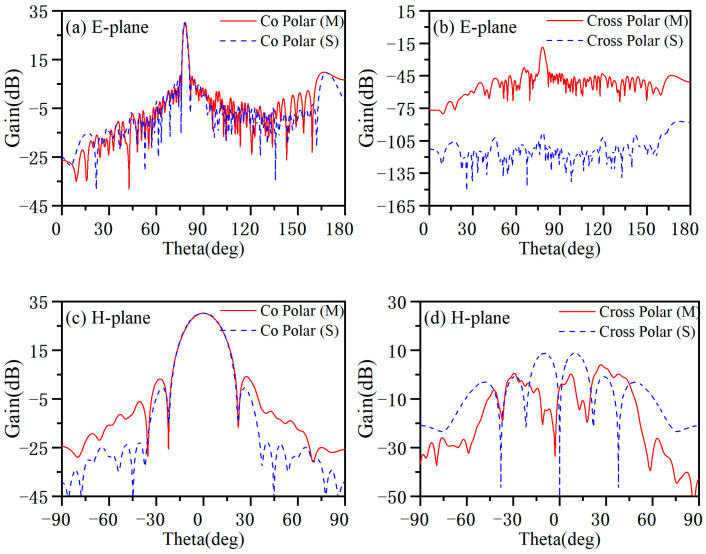
Measured and simulated radiation pattern of the antenna at 10 GHz. (**a**) Co-polarization radiation pattern of E-plane. (**b**) Cross-polarization radiation pattern of E-plane. (**c**) Co-polarization radiation pattern of H-plane. (**d**) Cross-polarization radiation pattern of H-plane.

**Figure 15 micromachines-14-02127-f015:**
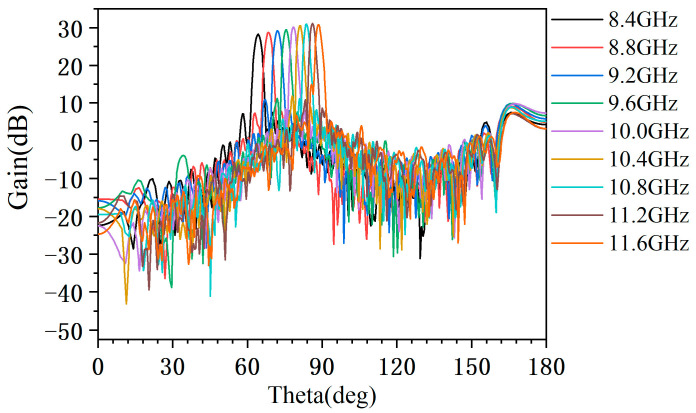
Measured radiation pattern of the antenna at 8.4–11.6 GHz.

**Figure 16 micromachines-14-02127-f016:**
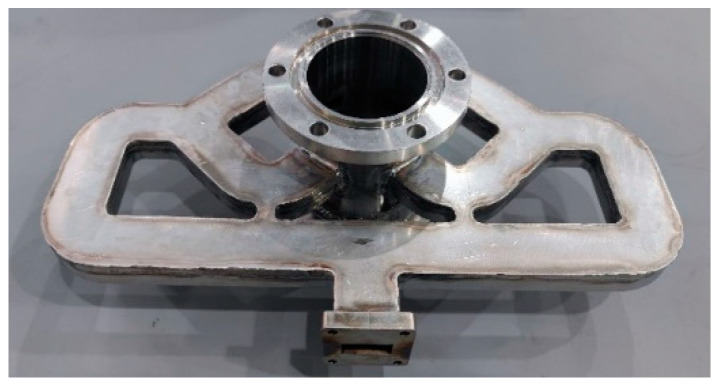
The structure of the coupler.

**Figure 17 micromachines-14-02127-f017:**
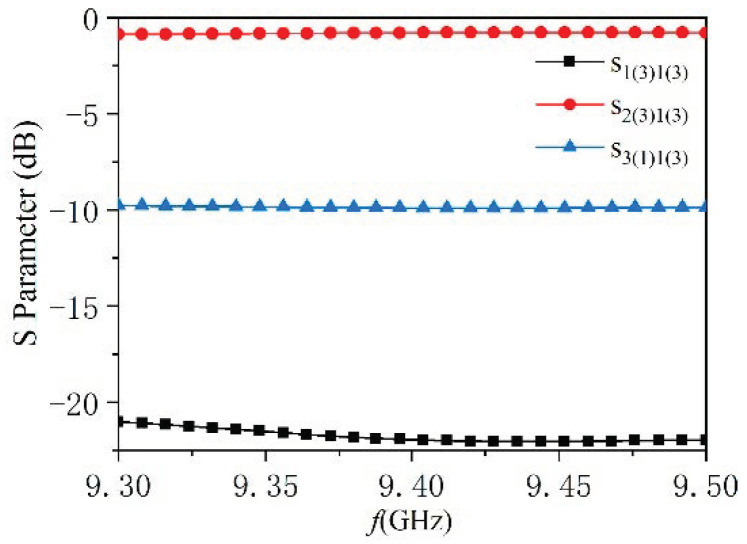
*S* parameters of the coupler.

**Figure 18 micromachines-14-02127-f018:**
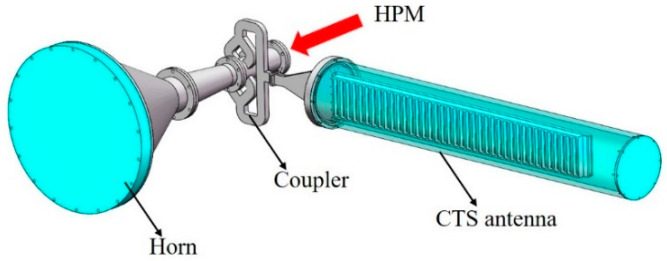
Schematic diagram of HPM injection experiment.

**Figure 19 micromachines-14-02127-f019:**
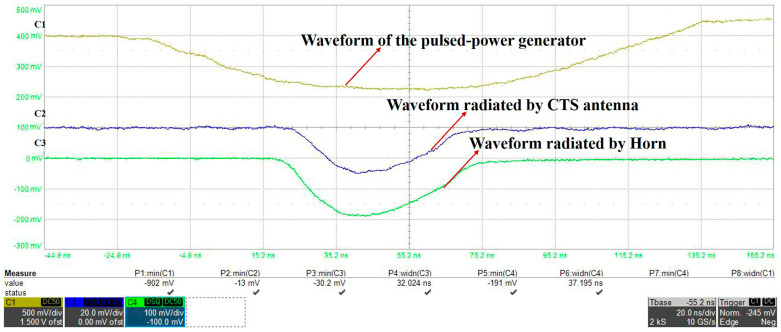
Typical waveforms received by an oscilloscope.

**Figure 20 micromachines-14-02127-f020:**
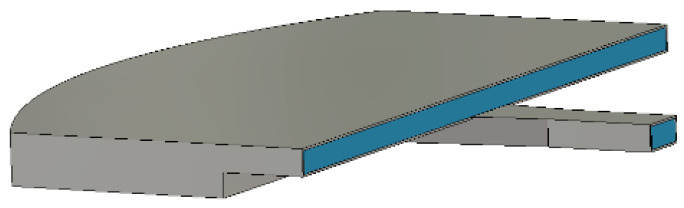
The feed pillbox box.

**Figure 21 micromachines-14-02127-f021:**
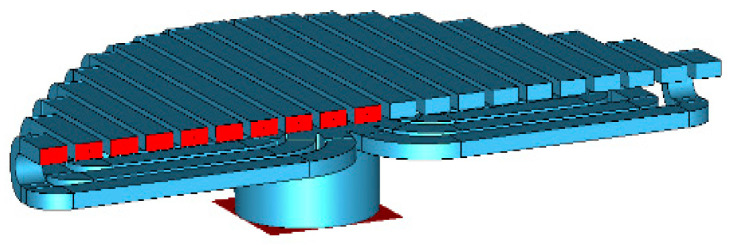
The feed network.

**Table 1 micromachines-14-02127-t001:** Antenna structure parameter.

Parameters	*h* _1_	*h* _2_	*h* _3_	*h* _4_	*h* _5_	*s* _2_	*s* _3_	*s* _4_	*s*	*t*	*d*
Values/mm	8.1	4.4	1.5	9.2	4	4	1.8	2.6	5.4	2.7	25

**Table 2 micromachines-14-02127-t002:** Antenna pattern simulation and measurement.

	Gain (dB)		SLL (dB)		3 dB Width (Deg)	
*f* (GHz)	Sim.	Mea.	Sim.	*f* (GHz)	Sim.	Mea.
8.4	28.5	28.2	−18.4	−20.7	2.5	2.5
8.8	28.7	28.8	−19.4	−19.7	2.4	2.4
9.2	29.2	29.2	−19.1	−18.2	2.3	2.3
9.6	29.6	29.4	−19	−18.3	2.2	2.2
10.0	30.1	30.2	−19.5	−20.4	2	2
10.4	30.6	30.5	−19.4	−18.8	1.9	1.9
10.8	31	31	−20	−19.9	1.8	1.8
11.2	31.3	31.1	−21	−20	1.7	1.8
11.6	31.3	30.8	−21.2	−16.8	1.8	1.9

## Data Availability

Data available in a publicly accessible repository.

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
