# Peer review of "Broadband Continuous Transverse Stub (CTS) Array Antenna for High-Power Applications"

_micromachines, 2023, doi:10.3390/mi14112127_

Round 1

Reviewer 1 Report

Comments and Suggestions for Authors

Considering in mind that steering  antennas are attracting great interest in the field of high-power microwave applications, the work presented here on the continuous transverse stub array antenna with wide bandwidth and high-power handling capacity may be significant in this field and represents an original approach. I recommend the publication of this work if some clarifications and improvements are made:

11)      There is a typo at the beginning of line 42.

22)      The authors have published some papers on high power microwave antennas in different journals, but they do not include them in the manuscript. These are the publications:

X. Zhao et al., "All-Metal Beam Steering Lens Antenna for High Power Microwave Applications," in IEEE Transactions on Antennas and Propagation, vol. 65, no. 12, pp. 7340-7344, Dec. 2017, doi: 10.1109/TAP.2017.2760366.

X. Zhao, C. Yuan, L. Liu, S. Peng, Q. Zhang and H. Zhou, "All-Metal Transmit-Array for Circular Polarization Design Using Rotated Cross-Slot Elements for High-Power Microwave Applications," in IEEE Transactions on Antennas and Propagation, vol. 65, no. 6, pp. 3253-3256, June 2017, doi: 10.1109/TAP.2017.2691460.

Y. Sun, F. Dang, C. Yuan, J. He, Q. Zhang and X. Zhao, "A Beam-Steerable Lens Antenna for Ku-Band High-Power Microwave Applications," in IEEE Transactions on Antennas and Propagation, vol. 68, no. 11, pp. 7580-7583, Nov. 2020, doi: 10.1109/TAP.2020.2979282.

L. Xu, S. Bi, J. Liu, C. Yuan, Q. Zhang and Y. Sun, "A Phase Synthesis Method for Reflectarray in High-Power Microwave Application," in IEEE Transactions on Plasma Science, vol. 50, no. 9, pp. 2858-2863, Sept. 2022, doi: 10.1109/TPS.2022.3199430.

The authors should also cite these papers in this manuscript and explain what is new in this manuscript compered to previously published papers from their side.

33)      The authors say that the efficiency of antenna can reach 98% without considering the ohmic loss and dielectric loss. They should provide at least an approximate estimate of the losses due to the resistance of the conductor and dielectric in their manuscript.

44)      Authors should measure the efficiency of the antenna or write how the efficiency of the antenna could be measured.

55)      The authors say that the matching layer is made of ABC material. They should explain what ABC material is.

66)      The authors should always use a space between the numerical value and the units. The space is missing in lines 116, 135 and 148.

77)      The visibility in Figure 19 is poor. The authors should improve the quality of Figure 19.

Author Response

Response to the Reviewer 1

Comments: Considering in mind that steering antennas are attracting great interest in the field of high-power microwave applications, the work presented here on the continuous transverse stub array antenna with wide bandwidth and high-power handling capacity may be significant in this field and represents an original approach. I recommend the publication of this work if some clarifications and improvements are made.

Response:

We would like to thank the respected reviewer in showing interest toward the proposed antenna and the insightful comments which helped us improving the quality of the manuscript. We have applied all the comments provided the respected reviewer. The answer to each comment is listed below.

Comment 1: There is a typo at the beginning of line 42

Response 1: We have corrected the error.

Comment 2: The authors have published some papers on high power microwave antennas in different journals, but they do not include them in the manuscript. These are the publications:

  1. Zhao et al., "All-Metal Beam Steering Lens Antenna for High Power Microwave Applications," in IEEE Transactions on Antennas and Propagation, vol. 65, no. 12, pp. 7340-7344, Dec. 2017, doi: 10.1109/TAP.2017.2760366.
  2. Zhao, C. Yuan, L. Liu, S. Peng, Q. Zhang and H. Zhou, "All-Metal Transmit-Array for Circular Polarization Design Using Rotated Cross-Slot Elements for High-Power Microwave Applications," in IEEE Transactions on Antennas and Propagation, vol. 65, no. 6, pp. 3253-3256, June 2017, doi: 10.1109/TAP.2017.2691460.
  3. Sun, F. Dang, C. Yuan, J. He, Q. Zhang and X. Zhao, "A Beam-Steerable Lens Antenna for Ku-Band High-Power Microwave Applications," in IEEE Transactions on Antennas and Propagation, vol. 68, no. 11, pp. 7580-7583, Nov. 2020, doi: 10.1109/TAP.2020.2979282.
  4. Xu, S. Bi, J. Liu, C. Yuan, Q. Zhang and Y. Sun, "A Phase Synthesis Method for Reflectarray in High-Power Microwave Application," in IEEE Transactions on Plasma Science, vol. 50, no. 9, pp. 2858-2863, Sept. 2022, doi: 10.1109/TPS.2022.3199430.

The authors should also cite these papers in this manuscript and explain what is new in this manuscript compered to previously published papers from their side.

Response 2: We have cited these papers in this manuscript and explain the differences and progress in the introduction.

Comment 3: The authors say that the efficiency of antenna can reach 98% without considering the ohmic loss and dielectric loss. They should provide at least an approximate estimate of the losses due to the resistance of the conductor and dielectric in their manuscript.

Response 3: The estimated ohmic loss is less than 1%, we have added it in the line 108.

Comment 4: Authors should measure the efficiency of the antenna or write how the efficiency of the antenna could be measured.

Response 4: In the simulation, we can get the S11 parameter and S21 parameter. Based on this, the efficiency of the antenna is got. In the measurement, an absorbing material is placed at the antenna end and the antenna only has one port, so we can only get the S11 parameter. If we want to get S21 parameter, the absorbing material should be moved and the S21 parameter can be measured. According to the simulation, the S21 parameter is less than -20 dB, the remaining energy is little. In fact, most of the antennas only have an input port, and only S11 parameter need to be measured.

Comment 5: The authors say that the matching layer is made of ABC material. They should explain what ABC material is.

Response 5: ABC material is the abbreviation of alloy blend composite, we have modified it in the line 129.

Comment 6: The authors should always use a space between the numerical value and the units. The space is missing in lines 116, 135 and 148

Response 6: We have checked the manuscript, and added a space between the numerical value and the units.

Comment 7: The visibility in Figure 19 is poor. The authors should improve the quality of Figure 19.

Response7: We have changed the Figure 19.

Reviewer 2 Report

Comments and Suggestions for Authors

This manuscript proposed a Continuous Transverse Stub Array antenna design for high power microwave (HPM) device. High power microwave devices have substantial applications in radar, long distance communication, satellite communication, and mobile communication. High-power handling capable antennas are an integral part of these HPM devices.  High power antenna research for HPM devices are crucial where power handling is an issue. In this regard, this manuscript could contribute significantly in modern day HPM devices.

However, several issues need to be resolved before this manuscript can be published.

1)      English need to be improved significantly. Several descriptions are cryptic. For an example, in the abstract, line 10, it should read, “which “wither improves or improved” the antenna operat-“.

2)      On the page 1, line 42, the authors wrote, “ Continumicromachines-2634319ous transverse”. It is not understood what the authors are trying to say.

3)      In the introduction section, the authors included several HPM devices. However, one of the significant HPM device, “Gyrotron” is not included. Please include Gyrotron and provide proper references.

4)      Simulation parameters, such as number of mesh cells, convergence criteria, and all the boundary conditions are important for this work. Please include all this information.

5)      It is not understood why the authors used magnetic boundary conditions on the sides instead of an open boundary condition.

6)      In the figure 7, it is not clear why the frontal lobe is tilted.

7)      Figure 11 shows a deep at 9.6 GHz for the measured data. Do you have any idea why there is a significant difference with the simulation result particularly for this frequency?

8)      In figure 19, x-axis is unreadable please update the graph.

9)      During the high-power test, a temperature measurement setup is crucial. However, no temperature data was shown. Please include an injected power vs. temperature graph.

10)   It desirable to describe clearly whether all the tests were carried out in CW or pulsed mode. The authors mentioned that pulse width was 35 ns. However, it not clear if all the tests were carried out in pulsed mode.

11)   -10 dB coupling coefficient have a room of improvement. Is there nay effort going on to improve the coupling coefficient?  

12)   What type of HPM source was used for the experiment?

Comments on the Quality of English Language

English writing must be improved. 

Author Response

Response to the Reviewer 2

Comments: This manuscript proposed a Continuous Transverse Stub Array antenna design for high power microwave (HPM) device. High power microwave devices have substantial applications in radar, long distance communication, satellite communication, and mobile communication. High-power handling capable antennas are an integral part of these HPM devices. High power antenna research for HPM devices is crucial where power handling is an issue. In this regard, this manuscript could contribute significantly in modern day HPM devices.

However, several issues need to be resolved before this manuscript can be published.

Response:

We would like to thank the respected reviewer in showing interest toward the proposed antenna and the insightful comments which helped us improving the quality of the manuscript. We have applied all the comments provided the respected reviewer. The answer to each comment is listed below.

Comment 1: English need to be improved significantly. Several descriptions are cryptic. For an example, in the abstract, line 10, it should read, “which “wither improves or improved” the antenna operat-“.

Response 1: We have made careful modifications and improved the English expression.

Comment 2: On the page 1, line 42, the authors wrote, “ Continumicromachines-2634319ous transverse”. It is not understood what the authors are trying to say.

Response 2: Dear reviewer, we do not find the problem you proposed, it may be a typography error.

Comment 3: In the introduction section, the authors included several HPM devices. However, one of the significant HPM device, “Gyrotron” is not included. Please include Gyrotron and provide proper references.

Response 3: The Gyrotron has been added in the introduction section, and several proper references has been provided.

Comment 4: Simulation parameters, such as number of mesh cells, convergence criteria, and all the boundary conditions are important for this work. Please include all this information.

Response 4: The magnetic boundary is set on the left and right sides to simulate the element. The open add space boundary is set on the top side of the element. The number of mesh cells of the element is about 87000.

Comment 5: It is not understood why the authors used magnetic boundary conditions on the sides instead of an open boundary condition.

Response 5: The input mode of the antenna should be TEM mode, so we used magnetic boundary conditions on the sides in the element’s simulation. But in practical application, the two sides are closed electrical walls and the fundamental mode inside the waveguide is TEn0. In our design, the width of the antenna is 121.5 mm, and the electric field transmitted in the waveguide can be considered as quasi-TEM mode. So, we do not use open boundary condition on the sides.

Comment 6: In the figure 7, it is not clear why the frontal lobe is tilted.

Response 6: In our design, the antenna is working at traveling wave state, the main lobe of the antenna is tilted in theory. If we want the main lobe is perpendicular to the antenna surface, the antenna should work at standing wave state, but the bandwidth is narrow.

Comment 7: Figure 11 shows a deep at 9.6 GHz for the measured data. Do you have any idea why there is a significant difference with the simulation result particularly for this frequency?

Response7: In our design, there is a lens in the feeding horn surface. The lens is made of polyethylene and alloy blend composite (ABC) matching layer for impedance matching. During the processing, due to limitations in the bonding process, the two materials cannot be completely bonded together which may cause the difference between the simulation result and the measured data.

Comment 8: In figure 19, x-axis is unreadable please update the graph.

Response8: We have changed the Figure 19.

Comment 9: During the high-power test, a temperature measurement setup is crucial. However, no temperature data was shown. Please include an injected power vs. temperature graph.

Response9: Although the power of the HPM is high, but the pulse width is only 35 ns, and the repetition is dozens of hertz, so the energy of the HPM is low. It will not cause the temperature of the antenna to increase. In HPM field, we do not need to carried out temperature measurement. In fact, we don't even need a cooling device.

Comment 10: It desirable to describe clearly whether all the tests were carried out in CW or pulsed mode. The authors mentioned that pulse width was 35 ns. However, it not clear if all the tests were carried out in pulsed mode.

Response10: The HPM is a pulse wave. All of our measurements are carried out under a power of GW level and a pulse width of 35 ns. We have clearly described in the paper.

Comment 11: -10 dB coupling coefficient have a room of improvement. Is there any effort going on to improve the coupling coefficient?

Response12: The coupling coefficient of the coupler can be improved. We are designing a coupler which coupling coefficient can be adjusted by change the insertion depth of a metal rod. It can achieve a wide range of coupling coefficient adjustment.

Comment 12: What type of HPM source was used for the experiment?

Response13: The HPM source is transit-time oscillator (TTO).

Round 2

Reviewer 2 Report

Comments and Suggestions for Authors

Thank you for answering most of questions.

However, few issues are still need to be resolved before this manuscript can be published.

1)      Simulation parameters, such as number of mesh cells, convergence criteria, and all the boundary conditions are important for this work. Please include all this information. The authors provided the boundary conditions and the number of meshcells. However, the convergence criteria are still not there. 87000 meshcells seems pretty low. Please comment.

2)      In the figure 7, it is not clear why the frontal lobe is tilted. It is not clear, how a tilted lobe would result in a broader bandwidth. Please describe it in the paper.

3)      During the high-power test, a temperature measurement setup is crucial. However, no temperature data was shown. The authors answered that a temperature measurement is not necessary. However, to convince the readers, a temperature vs. injected power graph or at least a peak power density approximation is desirable.

4)      Also, it is desirable to include all the updated information in the actual manuscript and not just the reply to the reviewers.

Comments on the Quality of English Language

Minor edits. 

Author Response

Comment 1: Simulation parameters, such as number of mesh cells, convergence criteria, and all the boundary conditions are important for this work. Please include all this information. The authors provided the boundary conditions and the number of meshcells. However, the convergence criteria are still not there. 87000 meshcells seems pretty low. Please comment.

Response 1: The calculation accuracy is set to -40dB, and adaptive mesh refinement has been used. The final number of mesh cells of the element is about 87000. The unit is small, so the number of the mesh cells can ensure that the calculation is accurate.

Comment 2: In the figure 7, it is not clear why the frontal lobe is tilted. It is not clear, how a tilted lobe would result in a broader bandwidth. Please describe it in the paper.

Response 2: The standing wave array can only operate at specific frequency, with slots spaced at half a waveguide wavelength, and its bandwidth is narrow. At the same time, the main lobe is perpendicular to the antenna surface. To obtain a wide bandwidth, the antenna must be designed as a traveling wave array, with a slot’s interval less than half a waveguide wavelength, and the beam direction of the antenna will deviate from the normal direction.

Comment 3: During the high-power test, a temperature measurement setup is crucial. However, no temperature data was shown. The authors answered that a temperature measurement is not necessary. However, to convince the readers, a temperature vs. injected power graph or at least a peak power density approximation is desirable.

Response 3: As I mentioned earlier, the temperature of the antenna will not change with the injection power because the energy of the HPM is too low, so we cannot provide a temperature vs. injected power graph. The temperature has never been the focus for HPM antennas. In fact, we are more concerned with the power handling capacity, which is also the key verification content of our high-power experiments. In addition, other researchers have not done any research on temperature and injection power in this field.

Comment 4: Also, it is desirable to include all the updated information in the actual manuscript and not just the reply to the reviewers.

Response 4: The revisions have been added in the manuscript and highlighted.